# Characterization of the Lipidome and Biophysical Properties of Membranes from High Five Insect Cells Expressing Mouse P-Glycoprotein

**DOI:** 10.3390/biom11030426

**Published:** 2021-03-14

**Authors:** Maria João Moreno, Patrícia Alexandra Teles Martins, Eva F. Bernardino, Biebele Abel, Suresh V. Ambudkar

**Affiliations:** 1Coimbra Chemistry Center, Chemistry Department, FCTUC, University of Coimbra, 3004-535 Coimbra, Portugal; patriciatelesmartins@gmail.com (P.A.T.M.); eva.fbernardino@gmail.com (E.F.B.); 2CNC—Center for Neuroscience and Cell Biology, University of Coimbra, 3004-535 Coimbra, Portugal; 3Laboratory of Cell Biology, CCR, National Cancer Institute, NIH, Bethesda, MD 20892, USA; biebeleabel@gmail.com (B.A.); ambudkar@mail.nih.gov (S.V.A.)

**Keywords:** lipidome, High Five insect cells, membrane proteins, biomembranes

## Abstract

The lipid composition of biomembranes influences the properties of the lipid bilayer and that of the proteins. In this study, the lipidome and the lipid/protein ratio of membranes from High Five™ insect cells overexpressing mouse P-glycoprotein was characterized. This provides a better understanding of the lipid environment in which P-glycoprotein is embedded, and thus of its functional and structural properties. The relative abundance of the distinct phospholipid classes and their acyl chain composition was characterized. A mass ratio of 0.57 ± 0.11 phospholipids to protein was obtained. Phosphatidylethanolamines are the most abundant phospholipids, followed by phosphatidylcholines. Membranes are also enriched in negatively charged lipids (phosphatidylserines, phosphatidylinositols and phosphatidylglycerols), and contain small amounts of sphingomyelins, ceramides and monoglycosilatedceramides. The most abundant acyl chains are monounsaturated, with significant amounts of saturated chains. The characterization of the phospholipids by HPLC-MS allowed identification of the combination of acyl chains, with palmitoyl-oleoyl being the most representative for all major phospholipid classes except for phosphatidylserines, which are mostly saturated. A mixture of POPE:POPC:POPS in the ratio 45:35:20 is proposed for the preparation of simple representative model membranes. The adequacy of the model membranes was further evaluated by characterizing their surface potential and fluidity.

## 1. Introduction

High Five™ insect cells are commonly used to overexpress membrane proteins, as is the case for the drug efflux transporter P-glycoprotein [1]. When expressed in mammalian cell lines, P-glycoprotein is fully glycosylated. On the other hand, P-glycoprotein expressed in High Five cells is not glycosylated. For this reason, ABC transporters including P-glycoprotein expressed in High Five cells are suitable for structural and biochemical/biophysical studies [2,3]. The functional properties of the membrane protein of interest are usually characterized while embedded in the original membranes [1], although some more specific properties and structural characterization requires the isolation of the protein of interest followed by purification and reconstitution in simpler model systems [4,5].

The lipid composition of the original membranes is very important to understand the results obtained, because membrane properties influence the structure, stability, and function of the membrane protein of interest [5,6,7,8,9,10]. Additionally, the active molecules interacting with membrane proteins are usually lipophilic and the lipid bilayer influences globally observed properties such as binding affinity and activity [8,9,10,11,12,13,14]. Since the relative affinity for the lipid bilayer and for the protein are not necessarily directly correlated, ignoring the mediator role of the lipid bilayer may lead to misinterpretations and to incorrect prediction of a protein’s specificity. The association of the active molecules with the lipid bilayer increases the local concentration in areas of the membrane surrounding the protein, orients amphiphilic molecules, and constrains their transversal location in the membrane, thus controlling their interaction with membrane proteins. 

The lipid composition is the most important property of the native membranes, controlling the global charge, thickness, and membrane fluidity [15,16,17,18,19,20,21]. The importance of the lipid fraction of biomembranes has often been overlooked, mostly due to difficulties in characterization. The extraordinary developments achieved in recent decades on gas chromatography with mass spectrometry detection (GC–MS) have made this important task not only possible but also simple and accurate [19,20,22,23,24,25,26]. This has led to the characterization of the lipid composition of a wide variety of biomembranes and the observation of significant variations between membranes obtained from different species and for cells grown under different conditions [15,16,17,18,19,22,23,26,27,28,29,30,31,32,33,34,35]. This shows the importance of characterizing the lipid composition of the native membranes in order to improve our understanding of the results obtained for the functional characterization of the proteins of interest when expressed in those systems.

In this work we characterized the lipidome and biophysical properties of membranes obtained from High Five insect cells. The major phospholipid classes were characterized by TLC, the fatty acids were analyzed by GC–MS, and the full characterization of the lipidome was performed by HPLC-MS. The membrane vesicles were also characterized by dynamic light scattering (DLS), zeta potential, and fluorescence anisotropy of the membrane probes DPH and NBD-C16.

We found the membrane lipid composition to be similar to that obtained previously for other insect cells, although significant variations were revealed. In addition, it was possible to obtain not only the overall acyl chain composition of the major lipid classes but also the characterization of the most abundant lipids. The lipidome characterization completed in this work includes all major lipid classes and minor components such as lysophospholipids, ceramides, and their glycosylates. A biophysical characterization was completed for total membranes and for representative lipid bilayers, allowing a better understanding of the role of the lipids in the overall properties of native membranes.

## 2. Materials and Methods

The lipids 1-palmitoyl-2-oleoyl-glycero-3-phosphocholine (POPC), 1-palmitoyl-2-oleoyl-sn-glycero-3-phosphoethanolamine (POPE), and 1-palmitoyl-2-oleoyl-sn-glycero-3-phospho-L-serine (POPS) were purchased from Avanti Polar Lipids, Inc. (Alabaster, AL, USA), the fluorescent probe 1,6-Diphenyl-1,3,5-hexatriene (DPH) was acquired from Sigma-Aldrich (Algés, Portugal), and N-hexadecyl-7-nitro-2,1,3-benzoxadiazol-4-amine (NBD-C16) was synthesized and purified as described before [36]. Reagents and solvents used were at the analytical grade or higher purity, and water was distilled and deionized with a final resistance ≥ 18 MΩ.

High Five insect cells (Invitrogen) were infected with recombinant baculovirus encoding (His)6-tagged mouse P-glycoprotein (p-gp). Membrane vesicles were prepared by hypotonic lysis of the insect cells, followed by ultracentrifugation to collect the membrane vesicles, as described in detail in the literature [1]. The membranes obtained were suspended in buffer (pH 7.5) containing 10% glycerol, aliquoted and stored at −80 °C until use.

The amount of protein in the membranes was quantified by the Sherman and Weismann method using Amido Black B dye (acquired to Alfa Aesar, Kandel, Germany) [37]. The lipids were extracted by the method described by Bligh and Dyer [38]. Briefly, CHCl_3_/MeOH 1:2 (*v/v*) was added to the sample (3.75 mL per mL of the sample in aqueous media), vortexed, and incubated on ice for 30 min. An additional volume of 1.25 mL CHCl_3_ and of 1.25 mL H_2_O were added and samples were vigorously vortexed. The mixture was then centrifuged at 180× *g* for 5 min at room temperature to obtain a two-phase system from which lipids were obtained. An aliquot of the total lipid extract was analyzed for total phosphorous quantification [39] to obtain the total amount of phospholipids in the sample. An average molar mass of 750 g was considered to calculate the phospholipid mass.

The lipid extract dissolved in CH_2_Cl_2_ was analyzed by TLC using silica gel plates to characterize the different phospholipid classes present. Prior to separation, the plates were washed with CHCl_3_/MeOH (1:1, *v/v*) and placed in an oven at 100 °C for 15 min to dry. Each plate was developed with CHCl_3_/EtOH/H_2_O/triethylamine (30:35:7:35 *v/v*), allowed to dry and sprayed with primuline (at 0.50 μg/mL in acetone/H_2_O (80:20)) and lipid spots were visualized under UV radiation (254 and 366 nm; Camag, Berlin, Germany). The spots were scraped off the plates and added to glass tubes for phosphorous quantification [39].

Fatty acid methyl esters (FAMEs) were prepared by reaction with a solution of KOH 2 M in MeOH, according to the methodology described previously [40,41]. Hexane solutions containing the FAMEs (0.75 μg) were analyzed by GC–MS on an Agilent Technologies 6890 N Network gas chromatograph (Santa Clara, CA, USA) equipped with a DB–FFAP column (30 m of length, 0.32 mm internal diameter, and 0.25 μm film thickness, 123-3232, J&W Scientific, Folsom, CA, USA). The GC equipment was connected to an Agilent 5973 Network Mass Selective Detector operating with an electron impact mode at 70 eV and scanning the range m/z 50–550 in a 1 s cycle in a full scan mode acquisition. The oven temperature was programmed from an initial temperature of 80 °C for 3 min; a linear increase to 160 °C at 25 °C min^−1^; a linear increase at 2 °C min^−1^ to 210 °C; and a linear increase at 30 °C min^−1^ to 250 °C followed by 10 min at this temperature. The injector and detector temperatures were 220 °C and 280 °C, respectively. Helium was used as a carrier gas at a flow rate of 1.4 mL min^−1^. The identification of each fatty acid (FA) was performed considering the retention times and comparison with MS spectra of FA standards (Supelco 37 Component Fame Mix, acquired to Sigma-Aldrich, Algés, Portugal) and those in the Wiley 275 library and AOCS Lipid Library. Internal standard methyl-nonadecanoate from Sigma-Aldrich was used (1 µg/mL). The relative amounts of FAs were calculated by the percent area method with proper normalization, considering the sum of all areas of the identified FAs. Results were expressed as means ± SD and in mass of the internal standard, expressed in micrograms per milligram of phospholipids.

Total lipid extracts were analyzed by hydrophilic interaction liquid chromatography on an Ultimate 3000 Dionex (Thermo Fisher Scientific, Bremen, Germany) high-performance liquid chromatography system (HPLC) with an autosampler coupled online to a Q-Exactive hybrid quadrupole mass spectrometer (Thermo Fisher, Scientific, Bremen, Germany). The mobile phase used consisted of a gradient starting at 10:90 to 90:10% *v/v* of eluents A:B where eluent A was H_2_O:ACN:MeOH at 1:2:1, and eluent B was ACN:MeOH at 3:2 (both containing 10 mM ammonium acetate). Each injection contained 5 μg of total lipid extract in 91 μL eluent B, and 4 μL of phospholipid standards mix (dMPC—0.02 µg, SM d18:1/17:0—0.02 µg, dMPE—0.02 µg, LPC—0.02 µg, dPPI—0.08 µg, dMPG—0.012 µg, dMPS—0.04 µg, dMPA—0.08 µg and CL—0.08 µg). The stationary phase was an Ascentis Si column HPLC Pore column (10 cm × 1 mm, 3 µm, Sigma-Aldrich), and elution was at a flow rate of 50 µL/min and 35 °C. 

Acquisition in the Orbitrap^®^ mass spectrometer was performed in both positive (electrospray voltage 3.0 kV) and negative (electrospray voltage −2.7 kV) modes, with a high resolution of 70,000 and AGC target of 1e6. Capillary temperature was 250 °C and the sheath gas flow was 15 U. For MS/MS determinations, a resolution of 17,500 and AGC target of 10^5^ was used and the cycles consisted of one full scan mass spectrum. Ten data-dependent MS/MS scans were repeated for each of the experiments, with a dynamic exclusion of 60 s and intensity threshold of 10^4^. Normalized collision energy™ (CE) ranged from 20 to 30 eV. Data acquisition was performed using the Xcalibur data system (V3.3, Thermo Fisher Scientific, Waltham, MA, USA). The facility of formation of positive and negatively charged ions depends on the structure of the lipid head group. The phospholipid classes, phosphatidylglycerols (PGs), phosphatidylinositols (PIs), phosphatidylserines (PSs) and cardiolipines (CLs), easily form negatively charged ions due to the loss of a proton from the polar head group, thus forming the anion [M − H]^−^. Phosphatidylethanolamines (PEs) may easily gain or lose a proton leading to [M + H]^+^ or [M − H]^−^ and the mass spectra was thus acquired in both the positive and negative modes. Phosphatidylcholines (PCs) were also analyzed in the positive and negative modes. The cation is produced from protonation of the phosphate and the charge in the choline group [M − H]^+^, while the anion is the result of an adduct formed between the lipid and acetate from the mobile phase [M + CH3COO]^−^.

The size of the membrane vesicles and their zeta potential was characterized using a Zetasizer Nano ZSP system (Malvern), and the fluorescence intensity and anisotropy was measured in a Cary Eclipse fluorescence spectrophotometer (Varian) equipped with a thermostatted multicell holder accessory and automated polarizers. The fluorescent probes (at a final concentration of 1 μM) were added from a 100 μM stock in DMSO to the membrane vesicles and to preformed vesicles of POPE:POPC:POPS 45:35:20 or pure POPC model membranes. The vesicles plus probe were allowed to equilibrate overnight at 37 °C with gentle stirring.

## 3. Results 

### 3.1. Lipid Composition of the Membranes Isolated from High Five Insect Cells

#### 3.1.1. Relative Abundance of Lipids and Proteins

The mass ratio of lipidic material to protein extracted from the membranes from High Five insect cells was 1.1 ± 0.2, of which 52% were phospholipids. This leads to a phospholipid to protein ratio equal to 0.57 ± 0.11 in the total membranes. Neutral lipids were also present in the membranes extract but they were not subjected to a detailed analysis. Small amounts of sterols have been previously found in similar cells [30,42], and were also identified by TLC in the membranes from High Five insect cells. Ceramides were also identified and characterized by HPLC-MS (see Section 3.1.5).

#### 3.1.2. Relative Abundance of the Different Phospholipid Classes

The different phospholipid classes were separated by TLC, identified against standards, and quantified by phosphorous analysis [39]. The results are shown in Table 1.

Phosphatidylethanolamines (PEs) were the most abundant, representing 45% of the total phospholipids in the membranes isolated from the High Five insect cells. The second most abundant class of phospholipids was phosphatidylcholines (PCs, 24%) followed by the negatively charged phosphatidylserines (PSs, 9%), phosphatidylinositols (PIs, 5%), and phosphatidylglycerols (PGs, 5%). Sphingomyelins (SMs) and lysophospholipids (LysoPLs) were also encountered in the membranes, at 5 and 1%, respectively. The isolated membranes also contain cardiolipins (CLs, 5%), reflecting some contamination by mitochondrial membranes.

#### 3.1.3. Quantification of the Fatty Acid Composition in the Phospholipid Pool

The overall composition of the fatty acids esterified with phospholipids is shown in Table 2. The most abundant is oleic acid (C18:1n9), representing almost half of the fatty acids content. Palmitoleic (C16:1n7), stearic (C18:0), and palmitic (C16:0) acyl chains are also abundant, representing 15–20% each. Small amounts of myristic (C14:0) and C18:1 other than oleic acid are also encountered, and minor quantities of very long fatty acids (arachidic C20:0, and eicosapentanoic C20:5). Monounsaturated fatty acids (MUFA) represented 66% of the total acyl chains, 33% were saturated (SFA) and only 1.4% were polyunsaturated (PUFA). 

#### 3.1.4. Lipidome Characterization of the Most Abundant Phospholipid Classes

The relative abundance of distinct acyl chains in the major phospholipid classes was characterized by HPLC-MS. The results obtained are shown in Figure 1, Figure 2 and Figure 3. 

The most abundant combination of acyl chains for phosphatidylethanolamines and phosphatidylcholines corresponded to C34:1 (Figure 1). Given the high abundance of oleic (C18:1) and palmitic (C16:0) acids, and the prevalence of SFA esterified with glycerol at carbon 1 and MUFA at carbon 2 [43,44], the major phospholipid species are thus identified as 1-palmitoyl-2-oleoyl-sn-glycero-3-phosphoethanolamine (POPE) and 1-palmitoyl-2-oleoyl-glycero-3-phosphocholine (POPC). Slightly shorter and longer acyl chains (32 and 36 carbons in both acyl chains) are also abundant in both phospholipid classes, and small amounts (below 3%) of even shorter and longer. As expected, very short acyl chains are mostly SFA while very long are highly unsaturated fatty acids (HUFA), in this way contributing to a relatively fluid membrane at the temperature of cell growth [29]. 

The lipidome of negatively charged phospholipids (PS, PI and PG) is shown in Figure 2. The results obtained for PI and PG were similar to those of PE and PC, with higher amounts of C34:1, and C32:1 and C36:1. However, large amounts are also observed of very long and highly unsaturated acyl chains (38:4–40:6). The acyl chain profile obtained for the PS lipid class is however very distinct from the others, with C38:0 being the most abundant combination of acyl chains.

A comparison of the distinct lipid classes is shown in Figure 3 with respect to the saturation of the acyl chains. The distinct profile of the PS lipids clearly stood out, with totally saturated chains being the most abundant, in contrast with the observation that a single unsaturation was usually found in the other lipid classes.

#### 3.1.5. Lipidome Characterization for the Additional Phospholipid Classes

In addition to the detailed characterization of the major lipid classes, some results were also obtained for other lipid classes found in the membranes at relatively small quantities. The results obtained are shown in Figure 4 and Figure 5. 

Sphingomyelin accounted for 5% of the total phospholipids, the large majority being 34:1 and 36:1 (Figure 4). The sphingosine moiety contributes with 18 carbons and a *trans* double bond. Thus, the most abundant fatty acids esterified with sphingosine are palmitoyl (C16:0) and stearoyl (C18:0), with small amounts of miristoyl (C14:0) and the longer acyl chain C20:0. Monounsaturated fatty acids (C14:1, C18:1 and C20:1) were also found, although in very small quantities (3% for the sum of all MUFA). Different ceramides could also be identified in the lipids extracted and the relative abundance of the different acyl chains was very similar to that observed for sphingomyelin. This similarity was expected, given that ceramides originate from sphingomyelins through the action of the enzyme sphingomyelinase. Monoglycosylceramides were also identified, with glucose, galactose or lactose attached to the ceramide moiety. A preference for ceramides with longer acyl chains is clearly observed. Lactosyl derivatives are much more abundant, with lactosyl ceramide C36:1 accounting for 50% of all glycosylceramides.

Lysolipids that originate from the major phospholipids present have also been identified; the relative abundance of their single acyl chain is represented in Figure 5. Lysolipids are usually obtained through the action of phospholipase A2, which cleaves the ester bond of the acyl chain at the *sn*-2 position on glycerol [45]. The acyl chain present in the lysolipids is therefore expected to be in the *sn*-1 position. The results obtained were in good agreement with those encountered for the respective phospholipid classes (Figure 1 and 2). The saturated acyl chain C16:0 is the most abundant on lysophosphatidylcholines (LPC), and supports the selection of POPC as the representative phosphatidylcholine for those membranes. The results obtained for lysophosphatidylethanolamine (LPE) show that C16:0 was also very abundant at the *sn*-1 position of glycerol from phosphatidylethanolamines, but C18:0 and C18:1 were found at high relative abundance. Globally the results were similar for LPC and LPE, with half of the acyl chains being saturated and half monounsaturated. Very long acyl and saturated acyl chains were obtained for lysophosphatidylserine (LPS), in agreement with the results obtained for this lipid class (Figure 2).

The cardiolipin (CL) class was also characterized, with C64:4 being the most abundant lipid (31% ± 5%), followed by C74:4 and C74:5 (33% ± 3% for the sum of both). Small amounts of the saturated phosphatidic acid with 28 carbons (C28:0) were also identified in the samples analyzed.

### 3.2. Biophysical Properties of the Membranes Obtained from High Five Insect Cells

#### 3.2.1. Surface Charge and Size of the Vesicles

The membranes obtained from High Five insect cells were stored in buffer containing Tris buffer pH = 7.5 and 10 w% glycerol, at a protein concentration of about 5 mg/mL. To decrease the turbidity of the solution and allow the characterization of their biophysical properties, they were diluted to 50 μg/mL in the same solvent. A typical correlogram obtained by DLS analysis is shown in Appendix A, and the size distribution obtained from the best fit is shown in Figure 6. As expected, a very broad distribution was obtained, indicating the presence of vesicles with diameters from 100 nm to above 1 μm (limited by the upper limit of the DLS equipment used). Fitting the correlograms with a single characteristic size (cumulant fits, Figure A1B in Appendix A) led to an average diameter of 956 ± 16 nm and a polydispersity index (PDI) close to 0.9 (the high PDI reflecting the broad distribution of vesicle sizes). Those numbers should be taken as indicative only because the best fit of the correlograms was of poor quality due to the polydispersity of the sample (Figure A1 in Appendix A).

The zeta potential of the vesicles was also characterized, being −14.4 ± 1.2 mV at the ionic strength of the suspension solvent (42.5 mM); see Figure A2 in Appendix A for the zeta potential distribution. The Gouy Chapman theory provides a way to calculate the surface potential and surface charge of the membranes [46,47,48]. Considering that the zeta potential corresponds to the potential at a distance of 0.2 nm from the membrane surface [49], the calculated surface potential was −16.5 ± 1.4 mV, corresponding to a surface charge density of −0.011 ± 0.001 C m^−2^. The lipid portion of these membranes contains a high fraction of negatively charged lipids corresponding to an average charge of −0.2 per phospholipid. This led to a predicted charge density of −0.033 C m^−2^ (considering an average area per lipid equal to 0.64 nm^2^) [50], which was three times higher than what was measured for the whole membranes. The discrepancy indicates that the proteins present in the membrane shield and/or partially compensate for the negative charges of the membrane lipids. This is supported by the observation that the whole lipid extract obtained from the membranes shows a significantly more negative surface potential than the total membranes, corresponding to a surface charge density twice as high as that obtained for the membranes.

#### 3.2.2. Membrane Polarity and Fluidity

Fluorescence anisotropy of the membrane probes DPH and NBD-C16 was used to characterize the dynamic properties of the membranes. This parameter depends on the mobility of the probes (higher mobility leading to smaller anisotropy), which in turn reflects the fluidity of their environment. The fluorophore of DPH is located in the non-polar interior of the lipid bilayer [51,52,53,54], thus reporting on the fluidity of this region of the membrane. For lipid bilayers in the gel phase, the acyl chains had very little mobility, leading to a DPH fluorescence anisotropy around 0.35. A large decrease in anisotropy is observed at temperatures above the lipid melting temperature [54], due to the increased mobility of the lipid acyl chains, which lead to a high fluidity in the region of the bilayer where DPH is located. On the other hand, the fluorophore of NBD-C16 is located at the membrane interface, [36,55,56] providing complementary information. The anisotropy was measured at 37 °C, which is the temperature at which the functional characterization of the membrane proteins is performed. The results are shown in Table 3, for the total membranes, for lipid bilayers with a simple but representative lipid composition (POPE:POPC:POPS 45:35:20), and lipid bilayers with POPC only.

The results obtained for the simple POPC membrane compare well with data reported in the literature, both for DPH [57] and for NBD-C16 [36]. The presence of POPE and POPS in the lipid bilayers led to a small increase in the fluorescence anisotropy of both probes (indicating a decrease in membrane fluidity). This result was expected, given the relatively higher temperature of the main phase transition for those lipids as compared to POPC (Tm(POPE) = 9 °C, Tm(POPS) = 14 °C, and Tm(POPC) = −2 °C) [58]. The anisotropy is still relatively small, as expected for membranes in the fluid state. A most significant increase is observed when the probes were inserted in Hi5 membranes, although the fluorescence anisotropy indicates that the membranes were also in a fluid state [54]. 

The fluorescence spectra of the probes inserted in the different membranes are presented in Figure 7. DPH shows the same spectra in all three membranes, indicating that the probe was in a media with similar polarity. This was expected from the location of the fluorophore in the non-polar portion of the membrane. A small blue shift was however observed for NBD-C16 inserted in the Hi5 membranes as compared with the pure lipid bilayers. This suggests that the membrane interface was less polar in the Hi5 membranes, the NBD fluorescent group being in a less hydrated environment [59]. The membrane proteins present in the membranes might explain the different properties of the membrane–water interface.

## 4. Discussion

The structural and functional properties of membrane proteins were strongly affected by their solvating lipids. This is particularly relevant for the efflux protein P-gp, for which some molecules may act as substrates or inhibitors depending on the environment where the protein is embedded [5]. Therefore, to interpret the protein functional properties in native membranes it is necessary to know their composition and properties. The interplay between the protein and lipid composition of biomembranes is still poorly understood, although some general relationships have been identified. This work represents a contribution to this goal.

As expected, the overall lipid composition of High Five insect cells overexpressing P-gp is similar to that reported previously for *Spodoptera frugiperda* (Sf9) insect cells and *Trichoplusia ni* (Tn) [29,30,42]. Significant differences are however observed, namely in the relative abundance of phospholipids and proteins, and in the relative proportion of the distinct phospholipids. A higher ratio of phospholipid to protein is observed (0.57 ± 0.11 as compared to 0.4 [42]), which is in the lower range of values encountered for plasma membranes, and higher than for mitochondrial and bacterial membranes [60,61]. The membranes characterized in this work are enriched in PEs, compensated by a smaller fraction of PCs. The fraction of negatively charged lipids is similar to that found previously in insect cells, but with different relative proportions of the distinct classes (PS > PG ≈ PI > PA). This is most likely due to difficulties in their separation [28,34]. In fact, the TLC separation performed in this work did not allow us to distinguish between PG and PA, and their quantification was only possible from the results obtained by HPLC-MS. Accordingly, more recent reports identify a large variety of negatively charged lipids, and PS stands out as the most abundant one [42].

The relative abundance of each acyl chain combination in the distinct lipid classes has not been previously reported for High Five or for any other insect cell line, being an important outcome from this work. We observe that the most abundant PEs and PCs were 34:1, 34:2 and 36:1, which is in close agreement with the higher abundance of C34 and C36 [42], and with the high percentage of C18:1, C16:1 and C18:0 [30]. From the overall acyl chain composition identified in previous reports, it was not however possible to predict class specific differences. The full lipidome characterization performed in this work shows that PIs were enriched in long and polyunsaturated acyl chains, while PSs show a predominance of unsaturated chains. The acyl chain composition of the different lipid classes may be a general property of the lipids, reflect cell type specificities, or be influenced by the membrane proteins. It is interesting to note that deviations from the average acyl chain compositions were mostly observed for negatively charged lipids, which are enriched in the annular fraction around P-gp [6,7,62].

In this work, some lipids present in the High Five membranes in low quantities were also characterized, including ceramides and their monoglycosylates (X-Cer). Interestingly, the acyl chain composition of X-Cer does not reflect that of the precursor, with a higher abundance of very long acyl chains. These long acyl chain ceramides cause interdigitation of the membrane leaflets with effects on the phase behavior [63] and with important consequences in cell signaling [64,65,66]. Elevated levels of X-Cer have also been associated with the development of multidrug resistance in cancer cells [67,68,69]. It would be relevant to study if overexpression of P-gp has some impact on the abundance and properties of X-Cer.

From the lipidome obtained with the membranes of Hi5 insect cells overexpressing P-gp, one may define the lipid composition for representative model lipid bilayers. This knowledge is very important to understand the properties of the membranes where the protein of interest was embedded, and to allow the preparation of proteoliposomes enriched in the protein of interest but with the lipid composition of the native membrane. It is also important to characterize the interaction of membrane protein substrates/inhibitors or modulators with the lipid portion of the membrane, allowing the discrimination between direct and membrane-mediated effects. As a first approximation, a representative model membrane may be prepared from POPE, POPC and POPS at the relative proportions of 45:35:20, where POPE and POPC are the most abundant lipids in each lipid class, and POPS represent all the negatively charged lipids in the membrane. The same acyl chain combination in all three lipid classes facilitates lipid mixing. More complex lipid compositions will preferably introduce variability in the acyl chain composition of the major lipid classes, for example using extracts from natural sources such as *E. coli* (enriched in PE and PG), Egg PC, and brain extract (enriched in PE, PC, PI, and PS).

In this work we also characterized the size, zeta potential, and fluidity of the Hi5 membrane vesicles, and this was compared with model membranes with the representative lipid composition. As expected from their lipid composition, the membranes were characterized by a negative surface potential. However, this was significantly less negative than predicted from the lipid composition of the membranes. This difference was partially explained by the dilution of the charged lipids in the Hi5 membranes due to the presence of neutral lipids and proteins. However, the difference observed might be the result of limitations in the method used to convert the zeta potential (at the slipping plane) into the surface potential. The 0.2 nm usually assumed between the surface and the slipping plane [49], may be a reasonable approach for pure lipid membranes, but not for protein containing membranes due to protein protrusion. This parameter should be re-evaluated for the case of protein containing membranes, to allow a more accurate characterization of their surface potential from the measured zeta potential.

The results obtained from the fluorescence anisotropy of the probes DPH and NBD-C16 show that the Hi5 membranes are in the fluid phase at 37 °C, as expected. Their fluidity was however lower than that of the lipid model membranes prepared from POPE:POPC:POPS (45.35:20). This reflected the presence of some additional lipids with longer and/or saturated acyl chains in the Hi5 membranes. The difference was however small, supporting the adequacy of this simple lipid mixture as a model for the lipid portion of the membranes obtained from High Five insect cells.

## Figures and Tables

**Figure 1 biomolecules-11-00426-f001:**
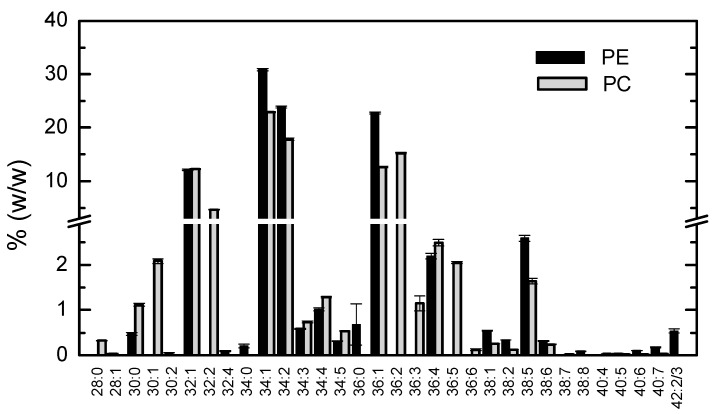
Relative abundance of the different acyl chains of phosphatidylethanolamine (PE, ■) and phosphadidylcholine (PC, ■) identified as [M + H]^+^. Results shown are the mean and standard deviation of 3 independent measurements from the same membrane isolation.

**Figure 2 biomolecules-11-00426-f002:**
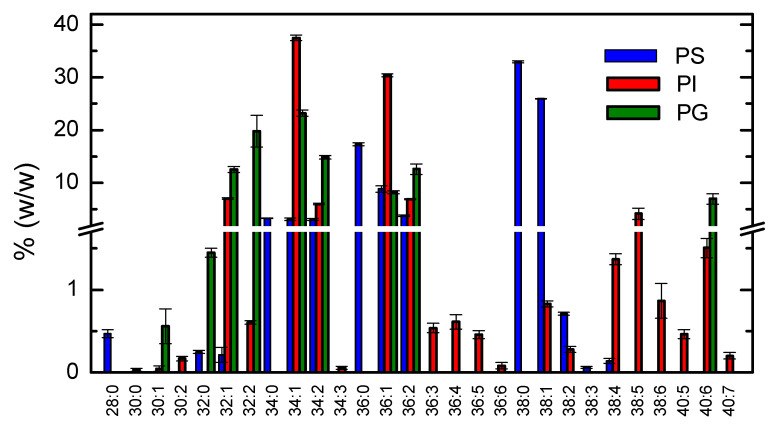
Relative abundance of the different acyl chains of phosphatidylserine (PS, ■), phosphatidylinositol (PI, ■), and phosphatidylglycerol (PG, ■) identified as [M-H]^-^. Results shown are the mean and standard deviation of 3 independent measurements from the same membrane isolation.

**Figure 3 biomolecules-11-00426-f003:**
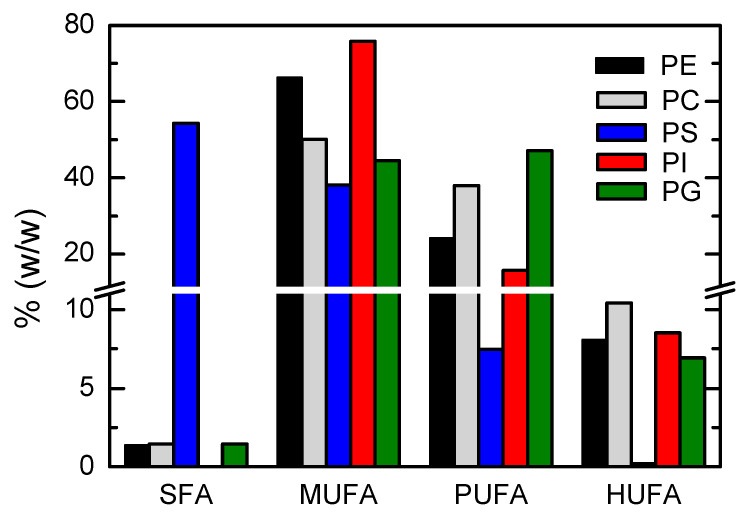
Relative abundance of saturated fatty acids (SFA), monounsaturated fatty acids (MUFA), polyunsaturated fatty acids (PUFA), and highly unsaturated fatty acids (HUFA) in the acyl chains of phosphatidylethanolamine (PE, ■), phosphatididylcholine (PC, ■), phosphatidylserine (PS, ■), phosphatidylinositol (PI, ■), and phosphatidylglycerol (PG, ■).

**Figure 4 biomolecules-11-00426-f004:**
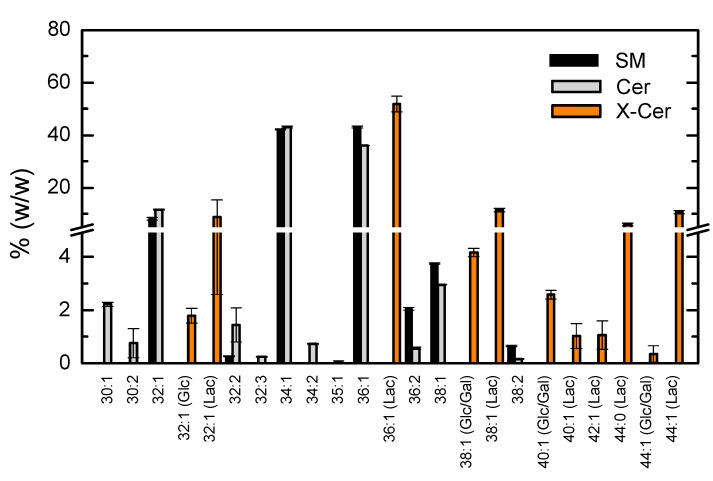
Relative abundance of the different acyl chains of sphingomyelins (SM, ■), ceramides (Cer, ■), and monoglycosylceramides (X-Cer, ■) identified as [M + H] + . Results shown are the mean and standard deviation of 3 independent measurements from the same membrane isolation.

**Figure 5 biomolecules-11-00426-f005:**
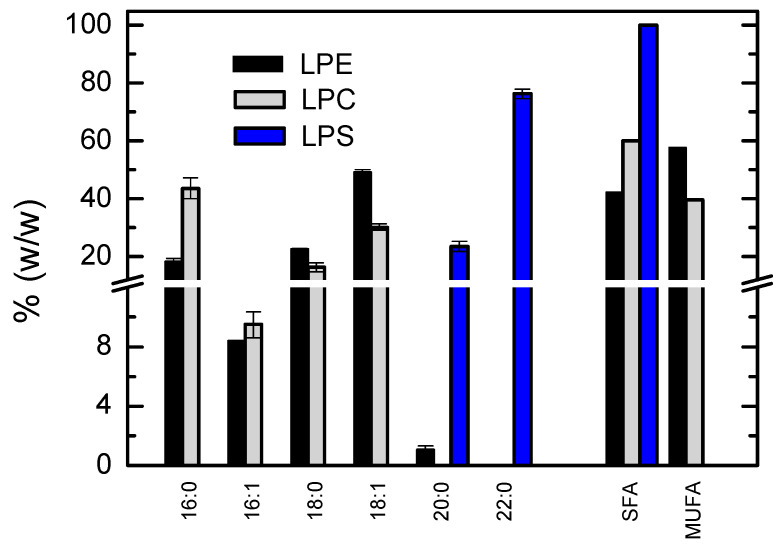
Relative abundance of the acyl chain of the lysolipids from phosphatidylethanomines (LPE, ■), phosphatidylcholines (LPC, ■), and phosphatidylserines (LPS, ■). Results shown are the mean and standard deviation of 3 independent measurements from the same membrane isolation.

**Figure 6 biomolecules-11-00426-f006:**
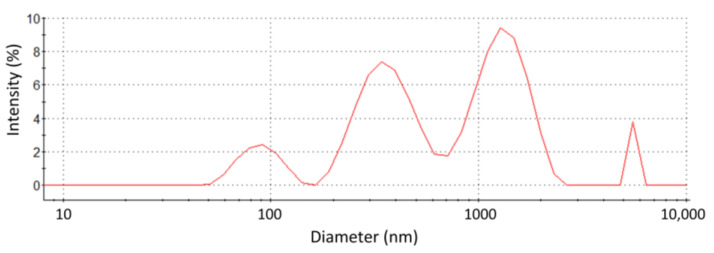
Size distribution of the membrane vesicles obtained from High Five insect cells. The sample was suspended in buffer (pH = 7.5) containing 10% glycerol at a lipid concentration of 50 μg/mL. The corresponding correlogram and best fit is shown in Appendix A.

**Figure 7 biomolecules-11-00426-f007:**
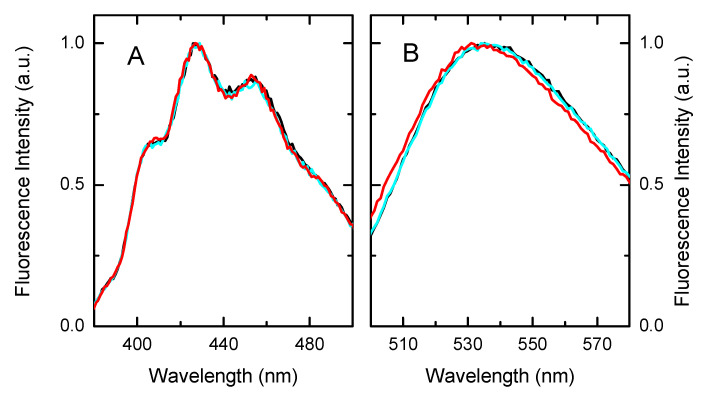
Fluorescence spectra of DPH (**A**) and NBD-C16 (**B**) inserted in lipid bilayers prepared from POPC (

), POPE:POPC:POPS in the ratio 45:35:20 (

), and in the membranes from High Five insect cells (

). The fluorescence intensity was normalized at the wavelength of maximum emission.

**Table 1 biomolecules-11-00426-t001:** Quantification of the different phospholipid classes found in membranes isolated from High Five insect cells.

Lipid Class	% (w/w) ^1^
CL	4.8
LysoPL	1.4
PC	24
PE	46
PG	5.1
PI	4.8
PS	8.7
SM	5.0

^1^ Assuming an average molar mass of 750 g for all phospholipids.

**Table 2 biomolecules-11-00426-t002:** Quantification of the most abundant acyl chains from the phospholipids pool of membranes isolated from High Five insect cells.

Acyl Chain	% (w/w) ^1^
C14:0	2.24 ± 0.27
C16:0	14.2 ± 0.16
C16:1n7	18.5 ± 0.14
C18:0	15.7 ± 0.33
C18:1n9	45.5 ± 1.4
C18:1	2.23 ± 0.34
C18:2	0.87 ± 0.25
C20:0	0.88 ± 0.18
C20:5	0.55 ± 0.06
SFA	33
MUFA	66
PUFA	1.4

^1^ Mean ± standard deviation of 3 independent measurements from the same membrane isolation.

**Table 3 biomolecules-11-00426-t003:** Fluorescence anisotropy of membranes isolated from High Five insect cells, and lipid bilayers prepared from a representative lipid composition and from POPC.

Fluorescent Probe	Membrane	Anisotropy
DPH	Hi5 membranes	0.17 ± 0.01
	PE:PC:PS 45:35:20	0.12 ± 0.01
	POPC	0.10 ± 0.01
NBD-C16	Hi5 membranes	0.25 ± 0.05
	PE:PC:PS 45:35:20	0.20 ± 0.04
	POPC	0.18 ± 0.03

## Data Availability

Original data will be provided when requested.

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
