# Peer review of "Characterization of the Lipidome and Biophysical Properties of Membranes from High Five Insect Cells Expressing Mouse P-Glycoprotein"

_biomolecules, 2021, doi:10.3390/biom11030426_

Round 1

Reviewer 1 Report

The work is interesting and well structured. The results are clearly described and well represented.
However, the discussion is difficult to interpret therefore it should be shortened in order to highlight the importance of the study.
For future studies it would be interesting to analyze the change in protein localization based on the lipid composition of the membranes.

Reviewer 2 Report

 The study "Characterization of the lipide and biophysical properties of membranes from High Five insect cells expressing mouse P-glycoprotein" by Moreno et al. is a careful study of the composition of a special cell type. The authors just report the data without any interpretation. They describe lipid chain composition, head group frequencies, lipid/protein ratios, charged lipid content, fluidity, ... of Hi5 cells in considerable detail. Finally, they propose an artificial lipid mixture that (in the opinion of the authors) will have similar properties as the native lipid membrane. I find this reasonable. The methods are well described. All of that seems sound and careful. The language is appropriate, the text is well written and it was an easy read for me. I find the results interesting and worthwhile being published. I would probably make use of these data once the paper is published. I have only minor comments/concerns.

Comments:

  1. I did not quite understand why this particular cell type that is over-expressing a mouse P-glycoprotein was chosen. Intuitively, I would have found it more interesting to see the composition of the native cells. The protein content might be higher in the chosen cell type, and the lipid composition different. What is the lipid/protein ratio in cells that do not over-express a particular protein?
  2. On page 6 the abbreviation HUFA was not introduced (it is somewhere later in the text)
  3. Page 9: Please say what the poly-dispersity index is and what it is a measure for. I was not familiar with this index.
  4. Page 10: Anisotropy is not defined. Indicate whether 0.17 for Hi5 membranes is more rigid (less fluid) than 0.1 for POPC. The whole point of "fluidity" could be better described.
  5. One thing that I found somewhat inconsistent is that the cells were grown at 27°C but the fluorescence anisotropy was measured at 37°C. As far as I know the lipid composition depends on growth temperature (at least in E. coli), and it would have been appropriate to measure the anisotropy at growth temperature. 37°C is probably just arbitrarily chosen because it is the "growth" temperature of mammals. The fluorescence anisotropy would strongly depend on temperature. Please justify the choice of growth and anisotropy measurement temperatures. Have you ever measured anisotropy at 27°C (or grown cells at 37°C)?
  6. Why is the protein content compared to the phospholipid content only and not to the total lipid content? I would find the latter number interesting and you probably have it in your files.

Summarizing, I find this a sound and complete work that only requires a few clarifications in the text.
